# Collaborating with Complementary and Alternative Medicine (CAM) Providers When Writing HPV Vaccine Review Articles

**DOI:** 10.3390/jcm9020592

**Published:** 2020-02-21

**Authors:** Michael J. Deml, Léna G. Dietrich, Bernhard Wingeier, Gisela Etter, Caesar Gallmann, Christoph Berger, L. Suzanne Suggs, Benedikt M. Huber, Philip E. Tarr

**Affiliations:** 1Swiss Tropical and Public Health Institute (Swiss TPH), 4051 Basel, Switzerland; michael.deml@swisstph.ch; 2University of Basel, 4051 Basel, Switzerland; lena.dietrich@unibas.ch; 3University Department of Medicine and Infectious Diseases Service, Kantonsspital Baselland, 4101 Bruderholz, Switzerland; 4Department of Pediatrics, Klinik Arlesheim, 4144 Arlesheim, Switzerland; Bernhard.Wingeier@klinik-arlesheim.ch; 5Association of Swiss Complementary Medicine Physicians Associations (UNION) and Private Practice, 8805 Richterswil, ZH, Switzerland; etter.praxis@bluewin.ch; 6Private Practice, 8804 Au, ZH, Switzerland; info@seepraxis.ch; 7Division of Infectious Diseases, University Children’s Hospital Zurich, University of Zurich, 8032 Zurich, Switzerland; christoph.berger@kispi.uzh.ch; 8Institute for Public Communication, Università della Svizzera italiana, 6900 Lugano, Switzerland; suzanne.suggs@usi.ch; 9Swiss School of Public Health, 8001 Zurich, Switzerland; 10Center for Integrative Pediatrics, Department of Pediatrics, Fribourg Hospital HFR, 1752 Fribourg, Switzerland; benedikt.huber@h-fr.ch; 11Faculty of Science and Medicine, University of Fribourg, 1700 Fribourg, Switzerland

**Keywords:** Switzerland, vaccine hesitancy, complementary and alternative medicine, vaccination communication, healthcare professionals

## Abstract

Novel strategies are needed to address vaccine hesitancy (VH), which correlates with complementary and alternative medicine (CAM). In Switzerland, CAM providers play important roles in vaccine counseling of vaccine hesitant (VH) parents, and traditional vaccination messaging tends to overlook CAM provider perspectives. In the setting of a Swiss national research program on VH, our key strategy has been to work together closely with CAM providers. To assess the feasibility of generating educational human papillomavirus (HPV) vaccine materials that would interest VH healthcare providers (HCPs), we invited four CAM providers to co-author two HPV vaccine review articles for general practitioners. We conducted thematic analysis of CAM provider comments to identify patterns that could complement and improve vaccination messaging from CAM perspectives. We identified several themes and generated an inventory of CAM provider messaging recommendations related to language use, presentation of background information, nuanced statements regarding HPV vaccine efficacy and safety, and communication tools that would be important to VH HCPs. Contrary to our initial expectations, and in an inclusive, respectful atmosphere of open dialogue, we were able to productively finalize our manuscripts. In the opinion of the CAM co-authors, the manuscripts effectively considered the communication needs and perspectives of VH HCPs. Engaging with CAM providers appears to be a feasible and innovative avenue for providing vaccine information and designing communication tools aimed at VH healthcare providers.

## 1. Introduction

In 2019, the World Health Organization (WHO) listed vaccine hesitancy (VH) as one of 10 major threats to global health [1]. Understanding VH remains a puzzling challenge for clinicians and public health authorities [2,3,4]. Strategies to address VH have included reinforcing traditional information approaches focused on improving knowledge [5], clinicians using presumptive rather than participatory communication styles [6], motivational interviewing [7], and vaccine mandates [8,9,10]. In response to the WHO’s announcement about the “threat” of VH, strategies to address VH are urgently needed.

Studies show a complex association between complementary and alternative medicine (CAM) use and VH [11], with popular narratives typically decrying CAM users and providers as categorically anti-vaccine [12]. However, research conducted in the setting of our national research program on VH [13,14] has suggested that providers of CAM in Switzerland are not categorically anti-vaccine. They prefer providing individualized vaccination counseling, which includes taking time to understand parent vaccine wishes, involving parents in vaccine decisions, and taking their vaccine concerns seriously [15].

Not only patients are vaccine hesitant; evidence now suggests the need to address VH among healthcare providers (HCPs) [16,17,18]. Based on repeated suggestions to us by Swiss CAM providers that VH HCPs are unlikely to even read review articles that employ the traditional communication approach that vaccines are safe and effective, we tested a strategy of listening to CAM providers’ perspectives and including them as co-authors in our review articles and communication tools for HPV vaccines. To our knowledge, CAM providers, who play an important role in vaccination counseling in Switzerland, have previously not been involved in similar collaborations. We hypothesized that, in order to reach VH HCPs, it is important to use nuanced language that considers particular vaccine concerns and communication needs. The endpoint of our collaboration was therefore to write HPV vaccine review articles together with CAM doctors so that VH HCPs would actually read them.

Here we investigate whether an innovative approach to engaging with CAM providers is feasible and productive. The objective of this article is to provide detailed characterizations of the insights and lessons we learned from including four Swiss CAM doctor comments, with two as co-authors, in two HPV vaccine review articles [19,20] written for Swiss general practitioners (both articles originally in German, one translated to French). Overall, our findings suggest that including CAM providers in the collaborative writing of vaccine review articles is a feasible, productive, and potentially promising approach to design communication tools for clinical practice in order to address VH.

## 2. Methods

In Switzerland, CAM use is prevalent among 25%–50% of the population [21,22], reimbursed through mandatory basic health insurance when provided by medical doctors with additional postgraduate training and certification in anthroposophic medicine, traditional Chinese medicine/acupuncture, phytotherapy, or homeopathy [23], and often provided by medical doctors who have undertaken additional CAM training [24]. According to the register of medical professions of the Swiss Federal Office of Public Health (FOPH), there are 1051 licensed medical doctors with additional accredited CAM training [25]. However, the number of licensed medical doctors who practice CAM or offer CAM services without full accreditation is likely higher. A representative study of pediatricians in Switzerland reported that 23% of pediatricians had attended CM training, but only 8% had a federal certificate in one or more of these methods [26]. The same study found that 97% of pediatrician respondents reported patients and parents inquiring about CAM, 65% were interested in pursuing CAM training, 16% provided CAM to their patients, and more than 50% used CAM for themselves or their families.

Despite vaccination being on a voluntary basis and the popularity of CAM in the Swiss context, vaccination rates for childhood vaccinations are overall relatively high for most routine childhood vaccinations (i.e., 87%–93% nationally for 2 doses of mumps, measles, and rubella vaccine for 2-, 8-, and 16-year-olds) [27]. However, there is high variance in HPV vaccination coverage between the different Swiss cantons (states). Cantonal coverage ranges from 19% to 79% for two doses of the HPV vaccine among 16-year-old females, and estimates for males are not yet available [27]. HPV vaccination is the vaccination that has achieved lowest coverage nationally among those recommended in the national vaccination plan [27,28].

We incorporated 4 trained CAM doctors’ comments and suggestions and included 2 of them as co-authors on 2 HPV vaccine review articles and communication tools that we wrote for general practitioners (GP) in Switzerland as the target audience [19,20]. One article was published, in both German and French, in Swiss Medical Forum, a peer-reviewed journal published by the Swiss Medical Association. In surveys, it has repeatedly been recorded as the most widely read continuing medical education journal in Switzerland, with 39,000 copies printed per issue [29]. The other article was published in German in Ars Medici, a popular Swiss family medicine journal, with 7500 copies printed per issue. Our goal was to write and frame review articles with CAM providers in order to craft messages about HPV vaccine which would be palatable also to VH HCPs while at the same time providing medically sound, evidence-based information. The intent of both articles was to generate tools for providers, including VH HCPs, to engage in high-quality HPV vaccine counseling, which has been demonstrated as an important factor when addressing patient VH [30,31,32].

In order to provide an inside look into the discussions with our CAM doctor co-authors, we assembled their suggestions and have subjected their comments to *thematic analysis*, a commonly used, pragmatic tool allowing researchers to identify and analyze patterns in qualitative data. We followed Braun and Clarke’s [33] suggestions regarding the 6 phases of thematic analysis: getting familiar with the data, generating initial codes, searching for themes, reviewing themes, defining and naming themes, and producing the report ([33], p. 87). We used the Framework Method [34] to structure our analyses and used MAXQDA software to code and organize our data into meaningful themes.

The 4 reviewing CAM doctors included 2 pediatricians specialized in anthroposophic medicine, a general practitioner (GP) specialized in homeopathy, and a GP of integrative family medicine. We did not assume that they provided suggestions representative of *all* CAM providers, but we included their contributions with the hypothesis that university researchers and public health authorities could learn from CAM provider’s experiences and perspectives when generating medical review articles and communication tools and vice versa. Of note, B.W. is a chief of pediatrics of the most renowned Swiss hospital specializing in anthroposophic medicine, B.H. is a chief of the first center for integrative pediatrics at a Swiss public hospital, C.G. is an influential private family medicine practitioner, and G.E. is president of the Swiss association of homeopathic physicians and of UNION of associations of Swiss physicians of complementary medicine.

## 3. Results

Through analysis of four CAM doctor recommendations for two manuscripts, we identified several patterns, which allowed us to group data into five overarching themes related to language and terminology, HPV vaccine efficacy, safety, communication tools for clinical consultation, and other important points deserving elaboration. The themes are presented in detail in the Tables where we display the specific content of CAM doctor recommendations, how we coded various sub-themes, and if we included their recommendations in the final manuscripts.

### 3.1. Language and Terminology

An overarching criticism was that the manuscripts appeared to be peddling pro-vaccination “propaganda”, which the CAM doctors argued would be unacceptable for VH HCPs. They hypothesized that prominent statements such as “HPV vaccine is safe and effective”, which already regularly appear in literature published by public health authorities and in medical guidelines [35], would lead VH HCPs to not even read the manuscripts. They recommended that we opt for nuances such as the vaccine “is considered” to be safe and effective. Other linguistic nuances included statements that were presented as proven facts, whereas there might be underlying uncertainty; they recommended we state, “most experts agree”, rather than “experts agree”.

We did not follow recommendations to remove two statements that CAM doctors considered to be “pro-vaccine propaganda”, related to vaccine efficacy in reducing genital warts, and waiting to vaccinate until 2020 until data on vaccine efficacy in preventing cervical cancer would be available. All agreed to underline that HPV vaccine today is well documented to reduce the incidence of cervical dysplasia (Table 1).

### 3.2. Vaccine Efficacy

The CAM doctors recommended we refine our discussion by highlighting that projections regarding vaccine efficacy to protect against anal and orophyngeal cancers were hypothetical at the time the articles were written in late 2017. Another item had to do with natural vs. vaccine-induced antibodies. They indicated that CAM providers are generally highly interested in the body’s ability to remove HPV and heal dysplasia after HPV acquisition by naturally induced immune responses. The CAM doctors agreed that vaccination induces HPV-specific antibodies much more reliably and at higher serum concentrations compared to natural infection, this being a salient argument in favor of HPV vaccine (Table 2).

### 3.3. Vaccine Safety

CAM doctors pointed out that our manuscripts unnecessary highlighted the local adverse reactions to the HPV vaccine. They encouraged us to instead focus on HPV-vaccine specific safety issues, such as case reports of rare but serious illnesses following vaccination, without inciting fear. While commending our emphasis on large-scale epidemiological evidence, acknowledging such case reports, they argued, would appeal to the tendency of CAM providers to incorporate “experiential” knowledge and anecdotal evidence into their medical practices [15,36]. Finally, CAM co-authors agreed that we point to the collective benefit of the HPV vaccine outweighing the minimal potential risks (Table 3).

### 3.4. Vaccine Communication Tools for Clinical Consultation

CAM doctors suggested we “inform patients about”, as opposed to “recommending” HPV vaccine. They also suggested that we prominently state that all vaccines remain voluntary in Switzerland and that trusting relationships can exist between patients and physicians regardless of patient vaccination decisions. Finally, they recommended that all patients be explicitly invited to voice concerns and ask questions about HPV vaccine, similar to informed consent prior to any operative procedure, even patients who state they wish to be vaccinated.

While CAM doctors pointed to how physicians can feel uneasy when broaching sexual matters, particularly with younger adolescents, they suggested we should not “de-sexualize” HPV vaccine. We did not follow their suggestions to include the possibility of waiting to vaccinate adolescents until age 15–16 years (when they would be more amenable to informed HPV vaccine discussions and decisions), which would not adhere to official recommendations to vaccinate at age 11–14 years, prior to adolescents becoming sexually active. Similarly, we did not include the suggested possibility of administering HPV and hepatitis B vaccines at separate times so as to attribute potential adverse effects to each individual vaccine (Table 4).

### 3.5. Additional Important Items

The following points seemed particularly pertinent because they drew our attention to items that might appeal to all readers, not just VH HCPs. For example, CAM doctors’ comments called our attention to the value of explicitly describing the sequence of events following natural HPV infection in order to help build a case for the importance of primary prevention against initial HPV infection via vaccine. Since VH is typically vaccine-specific, they encouraged us to provide more specific information on HPV infection before discussing HPV vaccine, that we elaborate on HPV vaccination for young men (since the recent recommendation to vaccinate boys and young men were likely not yet well known), and that we mention the cost of HPV vaccine, including how Swiss vaccination programs are funded, and that HPV vaccines are more expensive than other vaccines, and therefore likely lucrative for pharmaceutical companies. Finally, they agreed on mentioning HPV herd immunity thresholds in order to underscore the vaccine’s public health relevance (Table 5).

## 4. Discussion

Our collaboration with six CAM-oriented physicians has shown three main results. First, the process of generating HPV vaccine review articles and communication tools in collaboration with CAM providers appears feasible and may represent an important novel approach to better addressing VH. Second, we have generated an inventory of HPV vaccine messaging recommendations that are likely, according to four CAM doctors, to be relevant and palatable to VH HCPs but that may improve vaccine counseling by all HCPs. As shown in the results section, the messaging recommendations that would likely appeal to VH HCPs and CAM provider audiences dealt with being sensitive to language use, particularly language perceived as overtly and uncritically “pro-vaccine”, precisions about safety and efficacy data, emphasizing vaccination as a choice in clinical communication, and the inclusion of detailed information about HPV infections and HPV vaccine specific information. Table 6 (below) summarizes the key strategies recommended by CAM doctors for vaccine communication that gets vaccine-skeptical HCP on board. Third, and perhaps most importantly, our collaboration with four reviewing CAM doctors was constructive and resulted in messaging that was largely in favor of vaccination. This stands in contrast to popular narratives which pigeonhole CAM users and providers as categorically anti-vaccine [12], or those who express skepticism towards vaccination as ‘opponents’ [37].

Our collaborative approach is, to our knowledge, a first attempt of taking a CAM-inclusive rather than CAM-exclusive vaccine messaging approach. This may seem self-evident, but it goes beyond traditional top-down, knowledge deficit model efforts by (1) actively including VH HCPs in the target audience for HPV review articles, (2) listening to and engaging relevant stakeholders, (3) trying to understand the needs of the VH HCP audience, (4) targeting communication to those needs, and (5) designing culturally targeted interventions [38]. Thomson et al. point to the need to “understand and act upon the fact that there is no one-size fits all strategy to solve vaccine hesitancy and that collaborative efforts are needed and must be sustained over time” ([38], p. 6458). We were particularly impressed, despite initial skepticism on both sides, to be able to frame our reviews and communication tools in ways that were in line with official HPV vaccine recommendations, but also sensitive to providing HPV disease and vaccine information with nuanced language for issues where VH HCPs might have particular degrees of skepticism and concerns. Recognizing that language matters in healthcare [39,40], and that nuanced language choice can both empower and stigmatize in the oft-polarized discussions around vaccination, will be important for establishing meaningful dialogue in future efforts to address VH.

The aim of our collaboration with four CAM doctors in the design of HPV vaccine review articles and communication tools [19,20] was to ensure that these materials would be read not only by HCPs already favorable to vaccination, but useful particularly to the additional target audience of VH HCPs. To our knowledge, a CAM-inclusive approach has not yet been used in creating vaccine messaging materials for use in clinical practice. Additionally, this collaboration offers an innovative approach seeking to address VH around HPV vaccine, which is the vaccine listed on the Swiss vaccination schedule that has attained the lowest coverage in Switzerland [27,28]. Importantly, WHO cited healthcare professionals as “the most trusted advisor[s] and influencer[s] of vaccination decisions” [1]. Oehler [41] recently echoed these statements in response to the recent global measles resurgence: “New tools are needed for physicians and healthcare providers to reverse this trend and regain our role as patient’s best advocates”. In our view, our review articles clearly benefited from CAM doctors’ emphasis on patient-oriented medicine and from their focus on building trusting relationships with patients and families, which have been shown as important considerations for VH patients.

The work presented here suggests the feasibility of collaborative approaches that engage CAM providers, and we recommend such innovation to other researchers and for other health care issue and communication settings. For example, based on our productive cooperation, our research team is now conducting focus group discussions with CAM and biomedical providers in order to design vaccination communication interventions. We have now also successfully collaborated with the same CAM providers on other topics during the preparation of review articles for Swiss HCPs aiming at reducing antibiotic overuse in the setting of acute respiratory infections and asymptomatic bacteriuria and cystitis.

On a larger scale, our ongoing National Research Program on vaccine hesitancy [13,14] has recruited a network of more than 150 medical professionals throughout Switzerland, including more than 40 CAM practitioners. Given the popularity and interest towards CAM of the Swiss population [21,22] and from pediatricians [26], the incorporation of CAM perspectives and engagement with practitioners of both CAM and biomedicine into vaccination communication strategies is, in our view, a feasible, innovative, and reasonable approach to addressing VH in the Swiss context.

Our work has limitations. Even though the four collaborating CAM doctors were prominent CAM practitioners in their fields, they could not possibly represent the perspectives of all CAM providers in Switzerland, in particular the most hesitant providers. Rather, we took their recommendations by considering them as a sounding board for the creation of messaging that they felt would overcome the initial hurdle of rejection by VH HCPs and also gain traction within their various networks of CAM providers. However, we do not have data about the overall impact of the two HPV vaccine review articles among HCPs, nor do we have data about their influence on VH among HCPs. Such measurements go beyond the scope of this article.

Additionally, we did not integrate all CAM doctor suggestions into the final manuscripts in an ‘anything goes’ approach. Final decisions were negotiated internally with the conditions that we stay within the realms of evidence-based medicine, we adhere to official HPV vaccine guidelines, and that we make note of the quality of available evidence we presented in the final articles. This was possible in the vast majority of instances.

## 5. Conclusions

In this article, we show that our approach to engaging with CAM providers is feasible and productive. We have provided an inside look into an innovative approach for drafting HPV vaccine review articles and communication tools which seem relevant to all providers, including potentially VH HCPs. Using nuanced communication suggested to us by four CAM doctors, we were able to develop an intervention approach with the explicit goal of addressing VH among physicians and patients by improving the factual and communicative quality of HPV vaccine information. By inviting CAM doctors as co-authors, we sent a symbolically strong message to practitioners of CAM in Switzerland, indicating that we recognize their important clinical and public health role in providing vaccine counseling, particularly to VH individuals [15,42], and that biomedicine can incorporate CAM vaccination perspectives into materials destined for all HCPs. Through such collaboration, we have shown how it is possible to work with individuals that popular narratives would have us believe to be ‘the opponent’ when it comes to VH [37]. Addressing VH in clinical practice can, and should, benefit from multiple perspectives and involve all culturally relevant and appropriate stakeholders while informing medical review articles and communication tools for HCPs.

## Figures and Tables

**Table 1 jcm-09-00592-t001:** Language and terminology.

Item	Complementary and Alternative Medicine (CAM) Doctor Comments	CAM Co-Author Recommendations That We Included in Revised Manuscripts	CAM Co-Author Recommendations That We Did Not Include and Reasons for Not Including Them
Title	-The title (“HPV vaccine: Update 2018 for Clinical Practice”) is neutral and therefore acceptable. -When vaccine hesitant (VH) healthcare providers (HCPs) come across an article entitled “HPV vaccine: safe and effective”, [35] they are likely to not even read it because they will expect an excessively enthusiastic pro-vaccine article, the same “government propaganda” that they have seen over the past decades. -An advantage of your article in the eyes of VH HCPs will be that it is not written by authors affiliated with the Swiss Federal government.	-In the title, mention: -The aim of the manuscript is to provide well-balanced vaccine information. -The individual nature of vaccine decisions. -If you consider the statement that HPV vaccine is “safe and effective” to be essential, at least think about the nuance that HPV vaccine “is considered” by the authorities to be safe and effective.	
Statements that are pro-vaccine propaganda	-The CAM co-authors made comments on the following statements: “Genital warts have essentially disappeared in countries like Australia where high HPV vaccine coverage was achieved early on”. This might well be correct. However, it is propaganda because at the time of HPV vaccine introduction in Australia, this high level of vaccine efficacy was an unproven assumption. “Do not wait with recommending the vaccine to your patients until 2020, when data on prevention of cancer will be available”. This is correct. Today, however, this is “propaganda” because it is not yet known whether the vaccine prevents cervical cancer in addition to dysplasia.	-Such statements are not needed. Concentrate on the good pro-vaccine arguments that your article already has assembled. -Present readers with the data that are available today so they can reach their own conclusions.	-We did not remove this statement because we found the effect of HPV vaccine on genital warts to be relevant. -We did not remove the statement regarding data on cancer prevention by HPV vaccine becoming available in 2020 because it is factually accurate, and vaccinating against HPV will, in the meantime, lead to reductions of dysplasia cases.
Use of the term “protection”	-Do not use the term “protection” in an indiscriminate way. Sloppy use of language contributes to hesitancy towards vaccines and promotes the impression of “pro-vaccine propaganda”.	-Be careful to use specific terms for specific concepts. For example, be attentive to separate the following potential correlates of protection: -Documented doses of vaccine. -Induction of antibodies. long-term protection against dysplasia and/or cancer.	
General language use	-Your statement about “optimal” vaccine counseling makes us wonder about your intentions. -Is the goal to reach maximal vaccination coverage in the population? Or, is it, as CAM physicians will consider to be just as important, to help patients reach their personal, individually selected health and prevention goals?	-Be attentive to the formulation of the following statements: -HPV “can cause”, rather than “causes”, cancer. -HPV vaccination of boys “may prevent”, rather than “prevents”, cancer. -Prevention of genital warts via HPV vaccine “may be”, instead of “is”, worthwhile. -Use the word “reservations”, rather than “concerns”, about the HPV vaccine. -The data “suggests”, rather than “is solid”, that HPV vaccines are safe. -“Most experts agree”, rather than “experts agree”, that HPV vaccines are safe.	

**Table 2 jcm-09-00592-t002:** Vaccine Efficacy.

Item	CAM Doctor Comments	CAM Co-Author Recommendations That We Included into Revised Manuscripts
Protection against dysplasia vs. cancer	-Data on vaccine protection against cervical cancer is not yet available ^1^. This is an important limitation to government vaccine enthusiasm.	-Emphasize that published data only support protection against cervical dysplasia at present time and not against cancer. -Tell the readers when data on protection against cancer is expected. -Mention that no data yet exists to support vaccine efficacy against anal, oropharyngeal, and other cancers. -It is acceptable to state that experts expect vaccine to protect well also against anal and oropharyngeal cancer, even though data is not yet available. State this as a hypothesis.
Protection against dysplasia should not be underestimated	-It is helpful to mention that a diagnosis of cervical dysplasia can be associated with uncertainty and anxiety for many months.	-However, do not let this create the impression that the vaccine is 100% effective in preventing dysplasia, which could lead to reduced enthusiasm for continuing to undergo dysplasia screening at regular intervals. -Rather, mention that HPV vaccine and dysplasia screening are complementary cancer prevention methods for the time being.
Duration of vaccine protection	-The duration of protection is an important component of high-quality HPV vaccine counseling. -Hesitancy towards HPV vaccine is in part related to the fact that it is unclear if booster doses will be necessary.	-With only 13 years of experience with HPV vaccines, it seems premature to conclude that HPV protection is likely to last for at least 20 years. Use clear language about this, and provide references. -It seems premature to anticipate that vaccine protection will be lifelong. It is acceptable to state this as speculation.
Natural vs. vaccine-induced antibodies	-Your point that HPV vaccination induces HPV antibodies much more reliably, at higher serum concentration than “natural” HPV infection, and that vaccine-induced antibodies are protective against future infection with other HPV types, in contrast to naturally-induced antibodies, is important. -CAM providers are typically very interested in the concept that dysplasia is a sign that the body was unable to eliminate acute HPV infection, and that, in those with cancer, the body was unable to reverse dysplasia.	-Making such a statement is helpful to reduce HPV vaccine hesitancy among VH HCPs. -Making this point is also important because it distinguishes immunity towards HPV from immunity towards measles, where naturally acquired immunity persists lifelong whereas vaccine-induced immunity may wane and cease to protect. -The concept of the possibility of non-clearance of HPV infections and dysplasia, respectively, are important arguments for the HPV vaccine.

^1^ Note that these comments were made in late 2017 when cancer prevention data was not yet available.

**Table 3 jcm-09-00592-t003:** Vaccine Safety.

Item	CAM Doctor Comments	CAM Co-Author Recommendations That We Included into Revised Manuscripts
Local adverse effects of HPV vaccine	-You extensively discuss local adverse reactions to HPV vaccine, even though they are no different from other vaccines (i.e., pain, swelling at injection site).	-Do not let the discussion of local reactions create the impression that you are downplaying the importance of the long-term safety of HPV vaccine, which clearly is more important.
Case reports of serious adverse events following HPV vaccination	-There are case reports suggesting an association of HPV vaccine with auto-immune and other serious illnesses: The link between vaccines and their potential for triggering pre-existing autoimmunity is an exciting field of investigation and points to the individual immune response to any vaccine. -This is not unlike the well-known association of antibiotic treatment of otitis media with rare side effects (e.g., anaphylaxis to penicillin, clostridium difficile colitis). Nobody would put into doubt the need to use antibiotics, when they are indicated, due to safety concerns. Similarly, nobody would stop driving a car for safety reasons, even though there certainly are more deaths attributable to car accidents than to adverse vaccine effects. Being mindful about the rare possibility of a car accident likely leads to more careful driving. Being mindful about the rare possibility of a serious adverse vaccine reaction may lead to more careful medical care of HPV-vaccinated persons.	-Acknowledge the existence of case reports of rare but serious illnesses following HPV vaccination, without using fear. Giving these case reports some space will provide for a more well-balanced article and will be crucial to enhance the credibility and acceptability of your article to VH HCPs. -Even if they are, do not dismiss these case reports as irrelevant or inferior quality evidence. CAM providers are more likely than biomedical physicians to accept “experiential” types of evidence, such as personal experiences and individual case reports. Do not underestimate their importance. -These case reports should not be overly dramatized, as is currently the case in the media and via dramatic reports by overt anti-vaxxers.
Large scale epidemiological data suggests no serious long-term harm of HPV vaccine	-The overall statement that HPV vaccine is a safe vaccine is credible. -Most VH HCPs will agree that this large-scale epidemiological data is robust, high-quality evidence supporting the safety of HPV vaccine. WHO states that after 270 million doses of HPV vaccine administered, no signals of any serious vaccine adverse effects have emerged. This statement may be counterproductive and make VH HCPs more skeptical because these data have been collected, in part, by vaccine manufacturers. This reduces the trustworthiness of the data and, therefore, confidence in the safety of HPV vaccines.	-Consider putting these 2 concepts back to back in the manuscript by first stating, “Rare case reports suggest possible associations of HPV vaccine with multiple sclerosis”. -Follow this sentence by pointing out that large epidemiological studies “were unable to confirm these associations”. -Mention that long-term safety data is in being collected, in part, by vaccine manufacturers. -Even large-scale epidemiological studies cannot rule out that in rare cases, the HPV vaccine can be harmful to an individual patient. Mention this. -Therefore, HPV vaccine communication needs to be appropriately balanced and circumspect, particularly because there is no epidemiological emergency to vaccinate the population. Mention this.
Individual risk vs. collective benefit	Safety is a crucial aspect of any preventive measure, particularly for the HPV vaccine because it is now recommended for the entire population of young men and women.	-Mention that only a small fraction of vaccinated persons will individually benefit because HPV-associated cancers are overall rare. Mention vaccine safety as an important consideration because if everybody benefits from HPV vaccination, it is an acceptable assumption that the benefits outweigh the potential side effects. This will make the manuscript more balanced and acceptable to VH HCP readers.

**Table 4 jcm-09-00592-t004:** Vaccine communication tools for clinical consultation.

Item	CAM Doctor Comments	CAM Co-Author Recommendations That We Included into Revised Manuscripts	CAM Co-Author Recommendations That We Did Not Include and Reasons for Not Including Them
The decision to vaccinate or not to vaccinate	-Your statement that physicians should “patiently” and “extensively” provide HPV vaccine information is good. -Your statement that physicians should “inform” about HPV vaccine is good, and preferable to “recommending” the vaccine. -For those who agreed to receive HPV vaccine, proceeding to vaccinating without providing the patient with information is problematic (vaccine consent should be treated like pre-operative informed consent)	-Mention these points explicitly: -Vaccination remains voluntary in Switzerland. Doctors and patients are free in their decision to vaccinate or not. -Good health and a relationship of trust with physicians is possible for parents and adolescents with or without the HPV vaccine. -Patients have a right to be informed in a well-balanced, circumspect manner about HPV vaccine. -Physicians should accept and respect patients’ personal vaccination decisions. -Doctors should invite patients to voice their concerns and ask questions about HPV vaccine even if they want the vaccine and have no questions about it.	
Talking about sexual matters	-Many physicians feel uneasy when discussing sexual matters. This is a main reason for poor HPV vaccine counseling and contributes to HPV vaccine hesitancy. -Discussion of sexual topics requires an adequate amount of time. -Discussion of sexual topics with young adolescents in the recommended target age range (11–14 years old) makes physicians even more uneasy. -In general, vaccine counseling is inadequately reimbursed.	-Placing the emphasis on dysplasia/cancer prevention rather than prevention of an STD is acceptable. But do not “de-sexualize” the HPV vaccine. This is incompatible with an informed HPV vaccine decision. -Emphasize that HPV vaccination does not obviate the need for “safer sex” measures. -At 15–16 years, adolescents typically are ready for safer sex counseling. Many CAM physicians consider that HPV vaccine should not be given at 11–14 years but rather a few years later. Mention this as a possibility.	-We did not increase the HPV vaccine target age from 11–14 to 15–16 years and retained the official recommendation: HPV vaccine is ideally administered prior to individuals becoming sexually active.
Vaccine administration in conjunction with the hepatitis B vaccine	-It is correct to state that the HPV vaccine can be given at the same time as the hepatitis B vaccine.	-Some VH HCPs and CAM providers prefer to give HPV and hepatitis B vaccine at different times in order for them to be able to attribute potential safety issues to each individual vaccine. Mention this as a possibility.	-We did not modify our statement because we did not find HPV and HBV vaccine safety concerns sufficient enough to warrant this type of discussion.

**Table 5 jcm-09-00592-t005:** Additional important items.

Item	CAM Doctor Comments	CAM Co-Author Recommendations that We Included into Revised Manuscripts
Background information on HPV infection	-The manuscripts begin with pro-vaccine statements about the safety and efficacy of the HPV vaccine. This gives impression that the manuscript is “pro-vaccine propaganda”	-A well-balanced article should begin with extensive background information on HPV infection (natural history, complications), rather than HPV vaccine -Mention and underline that: -The recommendation to vaccinate boys and young men is not yet well known among general practitioners, rather than stating that we “now we need to introduce the vaccine” in young men. -HPV infection is the most common sexually transmitted infection. -HPV infection is harmless (i.e., transient) in >90% of the cases. -Cancer is a rare complication of HPV infections. -There are known risk factors (i.e., unprotected sex, number of sex partners) that support individualized HPV vaccine counseling. -There is no antiviral therapy available. This is in fact a pro-vaccine argument. -The goal of HPV vaccine is protection against dysplasia, rather than against a sexually transmitted infection.
HPV vaccine is different from other vaccines	-The concept that HPV vaccine is used to prevent dysplasia and cancer stands in contrast to all other vaccines where the goal is to prevent acute infections. -This point once more underscores the need to differentiate between individual vaccines when discussing the notion of “vaccine hesitancy”, particularly among patients who might not have categorical attitudes towards all vaccines. -This point is important also because, for example, not vaccinating against measles carries a considerable risk for the child to acquire measles, whereas not vaccinating against HPV carries little long-term risk, because most HPV infections are transient and can be prevented, at least partially, by other means (i.e., safe sex practices, fewer sexual partners, cervical cancer screening).	-This is a point definitely worth emphasizing because it is crucial for understanding why HPV vaccine might be important and therefore crucial for vaccine counseling. -Emphasize that there is no need to use fear as a communication strategy in order to promote the HPV vaccine. -It is OK to state that, similarly, there is no need by vaccine skeptics to use fear (of serious side effects, e.g., multiple sclerosis) to discourage vaccine.
Vaccination of boys and young men	-The discussion of cancer at the oropharynx and anus is important.	-This should be elaborated in order to address the prevalent notion that the HPV vaccine is only indicated for women.
High cost of the vaccine	-HPV vaccine is the most expensive, widely recommended vaccine. -The high vaccine cost puts a strain on already tight public health budgets, even in a rich country like Switzerland. -The HPV vaccine is very lucrative for vaccine manufacturers.	-The financial aspects surrounding HPV vaccine should not be kept secret, as has been the case until now. This secrecy is a major driver of vaccine hesitancy -The high cost of HPV vaccines should be discussed appropriately. Please clarify whether taxpayer money is being used to subsidize the vaccine.
HPV vaccine coverage of >70% may be sufficient to lead to herd protection in a given population	-This is an important point.	-Elaborate further with relevant references.

**Table 6 jcm-09-00592-t006:** Key general strategies recommended by 4 CAM doctors in order to improve vaccine communication and to get vaccine-skeptical health care providers on board.

Generate review articles on vaccines that are written by practicing doctors ○ i.e., authors that are independent of the health authorities and the federal government’s vaccine commission Consider that a significant proportion of health care providers are vaccine hesitant, not just patients. This is your target audience Begin the articles by providing detailed background information on the infectious disease aimed to be prevented by vaccination ○ Only in a 2nd step talk about the safety and efficacy of the individual vaccine(s) available Discuss immunity associated with natural infection and provide comparison with immunity associated with vaccination Avoid overtly pro-vaccine language ○ Avoid categorically repeating that all vaccines are safe and effective ○ Rather, emphasize the importance of individual patients reaching their personal, individually selected health and prevention goals. Vaccination remains a free, individual choice in countries that have no vaccine mandates in place, such as Switzerland Use precise and nuanced language ○ Avoid lumping all vaccines together. Very few patients are against “vaccination” in general ○ Clarify in each instance whether “protection” refers to protective antibody levels, prevention of persistent HPV infection, dysplasia, or cancer ○ Avoid stating that HPV vaccine protects against cancer before solid data is available ○ Speculation (on duration of protection after vaccination, on efficacy against cancer, etc.) is acceptable, but label it as such ○ For vaccines where herd immunity plays no or a limited role, avoid discussions of herd immunity and of a moral obligation to vaccinate Take vaccine safety concerns seriously ○ Acknowledge the existence of rare cases of serious illnesses following vaccination ○ In a second step, mention robust, large scale epidemiological vaccine safety data Discuss potential problems and side effects from the vaccination, discuss critical papers, alternatives and/or the potential consequences of vaccination and non-vaccination Emphasize the importance of the provider investing time and effort in order for the patient to be able to reach an informed vaccination decision Mention the price and financial implications of each vaccine ○ Be aware of one of the major determinants of vaccine hesitancy: the prevalent perception that the health authorities are collaborating with and/or influenced by vaccine manufacturers

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
