# Peer review of "Collaborating with Complementary and Alternative Medicine (CAM) Providers When Writing HPV Vaccine Review Articles"

_jcm, 2020, doi:10.3390/jcm9020592_

Round 1

Reviewer 1 Report

This manuscript describes a unique approach to learning more about the HPV vaccine-related views and perspectives of complementary and alternative medicines providers - and importantly, to learn whether it is possible to use those views to increase the value of HPV vaccine-related education materials to CAM providers. Given that CAM use is prevalent among 25-50% of the population in Switzerland, it seems quite important to find ways to productively engage with CAM providers on vaccines and vaccination recommendations. From this perspective, the effort undertaken and described here is very helpful. The manuscript does a very good job of describing the rationale for the effort and in describing, via the tables, the views of CAM physicians and whether/how those views could be utilized in the manuscripts' content. A major/significant weakness of the manuscript is the small number of CAM providers who were engaged in the effort, which means many readers would likely question the scientific soundness of the findings. I believe the authors have done a good job of addressing major concerns raised in the first round of reviews but I think it may be necessary, given that greater involvement of CAM providers is not possible at this point, to improve the framing of this effort. I would suggest, for instance, reconsidering the title, which at present implies that this effort addressed vaccine hesitancy. It is not possible given what was done, to know whether the input from the four CAM providers impacted vaccine hesitancy, including HPV vaccine hesitancy. Thus, I would recommend a title more reflective of this effort's purpose - which was to 1) see whether it was feasible and productive to engage with CAM healthcare providers on two HPV vaccine review articles and 2) learn if such engagement would produce helpful editorial suggestions (e.g., comments that would result in manuscripts that would help CAM providers better understand HPV vaccine and HPV vaccination recommendations). Along with revising the title, this manuscript would also likely benefit from focusing more on the approach and engagement and less on framing this project as "research" or "science." In fact, at present the Abstract does a very good job of taking such an approach. It may be helpful in both the Introduction and Conclusion to note this project was about seeing whether an innovative approach to engaging with CAM providers was feasible and productive - and not a scientific study, per se. I recommend this because I see much value in the effort described in this study and the lessons learned, but I also recognize that this effort would have major weaknesses if viewed as a scientific study.

Author Response

Our responses to the reviewer's specific comments below in bold. 

This manuscript describes a unique approach to learning more about the HPV vaccine-related views and perspectives of complementary and alternative medicines providers - and importantly, to learn whether it is possible to use those views to increase the value of HPV vaccine-related education materials to CAM providers. Given that CAM use is prevalent among 25-50% of the population in Switzerland, it seems quite important to find ways to productively engage with CAM providers on vaccines and vaccination recommendations. From this perspective, the effort undertaken and described here is very helpful. The manuscript does a very good job of describing the rationale for the effort and in describing, via the tables, the views of CAM physicians and whether/how those views could be utilized in the manuscripts' content.

A major/significant weakness of the manuscript is the small number of CAM providers who were engaged in the effort, which means many readers would likely question the scientific soundness of the findings. I believe the authors have done a good job of addressing major concerns raised in the first round of reviews but I think it may be necessary, given that greater involvement of CAM providers is not possible at this point, to improve the framing of this effort.

I would suggest, for instance, reconsidering the title, which at present implies that this effort addressed vaccine hesitancy. It is not possible given what was done, to know whether the input from the four CAM providers impacted vaccine hesitancy, including HPV vaccine hesitancy. Thus, I would recommend a title more reflective of this effort's purpose - which was to 1) see whether it was feasible and productive to engage with CAM healthcare providers on two HPV vaccine review articles and 2) learn if such engagement would produce helpful editorial suggestions (e.g., comments that would result in manuscripts that would help CAM providers better understand HPV vaccine and HPV vaccination recommendations).

Response: We agree and have changed the title to “Collaborating with complementary and alternative medicine (CAM) providers when writing HPV vaccine review articles” in order to clarify. To our knowledge, ours is the first attempt to undertake a CAM-inclusive approach when writing review articles about vaccination. This is a novel and innovative approach. This information is already included in lines 226-227. In addition, as stated in lines 131-135, the CAM co-authors are in fact leaders in their respective fields in Switzerland.

Along with revising the title, this manuscript would also likely benefit from focusing more on the approach and engagement and less on framing this project as "research" or "science." In fact, at present the Abstract does a very good job of taking such an approach. It may be helpful in both the Introduction and Conclusion to note this project was about seeing whether an innovative approach to engaging with CAM providers was feasible and productive - and not a scientific study, per se. I recommend this because I see much value in the effort described in this study and the lessons learned, but I also recognize that this effort would have major weaknesses if viewed as a scientific study.

Response: We agree and have included a sentence as suggested by the reviewer in both Introduction (lines 75-76) and conclusion section (lines 289-290).

Reviewer 2 Report

This is an interesting and novel approach that could really have an impact.

I appreciate the clearly structured presentation of the results.

Yet, I feel that the discussion section could be more substantial. I am particularly wondering if it would be possible to condense the requests of the of the CAM providers on a meta level. Are there, for example, general traits of these requests that could be summarized into key strategies for getting CAM providers on board?

Or, else, did the cooperation of CAMs only entail vaguer formulations, like suggestions instead of recommandations etc.? Then, their impact would be considerably lower than one could have expected. In fact, the mere weakening of straigth forward formulations wouldn't hurt anybody and could be seen as an easy strategy to just undermine CAMs' opposition.

Moreover, it seems to me, that more concrete and substantial CAM requests, like, for example, to postpone HPV vaccination to older age of adolecscents, were not taken into account by the decision makers. Did the CAMs accept this without protest? Why? Did they, eventually, subordinate to the official vaccination policy in order to stay in the game? Was, therefore, their role inferior to the role of the biomedical stakeholders, and if so - why?

These are some examples of questions I would like to be discussed in the discussion and conclusion section.

Author Response

Our responses to the reviewer's comments are below in bold. 

This is an interesting and novel approach that could really have an impact.

I appreciate the clearly structured presentation of the results.

Yet, I feel that the discussion section could be more substantial. I am particularly wondering if it would be possible to condense the requests of the of the CAM providers on a meta level. Are there, for example, general traits of these requests that could be summarized into key strategies for getting CAM providers on board?

Response: We agree and have added an additional sentence about this (lines 214-218) in the Discussion section. Regarding the question of getting CAM providers on board, we have included a sentence about how such an invitation to CAM doctors as co-authors “sent a symbolically strong message to practitioners of CAM in Switzerland” (lines 294-297) in the hope that this would promote wider readership of the review articles. Finally, we agree and have added a Table 6 that summarizes key strategies for getting CAM providers on board in the discussion.

Or, else, did the cooperation of CAMs only entail vaguer formulations, like suggestions instead of recommandations etc.? Then, their impact would be considerably lower than one could have expected. In fact, the mere weakening of straigth forward formulations wouldn't hurt anybody and could be seen as an easy strategy to just undermine CAMs' opposition.

Moreover, it seems to me, that more concrete and substantial CAM requests, like, for example, to postpone HPV vaccination to older age of adolecscents, were not taken into account by the decision makers. Did the CAMs accept this without protest? Why? Did they, eventually, subordinate to the official vaccination policy in order to stay in the game? Was, therefore, their role inferior to the role of the biomedical stakeholders, and if so - why?

Response: We have discussed these issues in lines 283--287. The CAM co-authors agreed with the conditions that we adhere to official vaccination recommendations and stay within the realms of evidence-based medicine in our review articles in final publication results.

These are some examples of questions I would like to be discussed in the discussion and conclusion section.

Reviewer 3 Report

The abbreviation HPC in line 32 hasn't been previously explained in the abstract.

In lines 129 - 133, consider changes in order to avoid identifying specific individuals.

Author Response

Our comments to the reviewer's comments are included below in bold. 

The abbreviation HPC in line 32 hasn't been previously explained in the abstract.

Response: We have modified this to explain the abbreviation.

In lines 129 - 133, consider changes in order to avoid identifying specific individuals

Response: The co-authors on this paper have agreed with their initials being presented here in the article. We are open to removing the initials and leave this open for an editorial decision.

This manuscript is a resubmission of an earlier submission. The following is a list of the peer review reports and author responses from that submission.

Round 1

Reviewer 1 Report

This is a well conceived and designed research project and the manuscript is well organized and written. It would help, if possible, to provide more specific information regarding vaccination rates in Switzerland, particularly for HPV vaccination given that is the focus here. It would also help, if possible, to provide more data regarding the overall numbers of traditional/mainstream physicians in Switzerland and CAM providers in Switzerland. In the Discussion, it would also be helpful if the authors addressed whether/how feasible they believed it would be to undertake efforts that tried to engage more CAM providers on vaccine education efforts. It is one thing to engage with four people, but thinking longer term and more broadly, would it be feasible or even possible to have greater involvement and engagement - or would factors like lack of tolerance/acceptance by mainstream providers or fear of being coopted by CAM providers prove to be substantial barriers? 

Originality/Novelty: This study and manuscript are very original and well defined. The authors have done an excellent job of describing the uniqueness of the research approach, with the approach applied to the important issue of finding new ways to address vaccination hesitancy. Importantly, the study is focused on HPV vaccine and on healthcare providers. The results clearly provide an advance in current knowledge, both with respect to the methods used (i.e., engagement of complementary and alternative medicine providers) and findings (i.e., that input from CAM providers can be productively used to shape the educational messages and materials used to reduce HVP vaccination hesitancy).
Significance: The results of this study were well illustrated using five tables and appropriately interpreted. The tables were an excellent way to illustrate CAM providers’ comments and suggestions and how they were utilized. The authors did an outstanding job of describing how the comments and suggestions differed from those typically used in government produced HPV education materials.
Quality of presentation: This was a very well structured and written manuscript, with excellent use of tables to highlight findings and their application. The data and analyses were presented appropriately, and the authors exercised sound judgment regarding the value and application of qualitative research findings.
Scientific soundness: While this was a study involving qualitative methods and data, the authors were consistently cognizant of the value and limits of their methods and data. They did a very good job of explaining the utility of the approach, the limits of the approach, and showing how engagement of even a relatively small number of CAM providers could improve the framing and messaging related to HPV vaccination education efforts, particularly those directed at a larger group of CAM providers. Importantly, as a result of the tables and methods description provided, other researchers would be able to reproduce the results/
Interest to the Readers: The overall research approach as well as the conclusions should be quite interesting to Journal readership. Given CAM providers are often seen as contributing to vaccination hesitancy, there should be much interest in this study because 1) it illustrates a novel and potentially productive approach to engaging with CAM providers on vaccination education materials and efforts and 2) it shows contributions, challenges, and limits of such engagement. It is also likely this study will attract interest because it also shows that CAM providers 1) often appear to hold mainstream or traditional medicine and science to different (e.g., higher) evidentiary standards than they do for CAM approaches and 2) CAM providers have assumptions and biases toward vaccines and vaccination that may not be grounded in reality, such as equating the cost of a vaccine with its profitability, including by apparently ignoring or discounting the significant investments vaccine manufacturers need to make in clinical trials to assess safety and efficacy and then in manufacturing capacity. There is thus much in this paper that should be of interest to a wide number of people.
Overall Merit: There is significant overall benefit to publishing this work, including because it uses and describes a novel approach to productively engaging with healthcare providers who are often not strongly supportive of vaccines and vaccination. Importantly, it identifies a way to productively engage with CAM providers as well as approaches that may reduce CAM providers’ vaccine hesitancy.
English Level: The English language used in the manuscript is appropriate and very understandable. This is a well-structured and written manuscript.

Reviewer 2 Report

The authors want to provide the contribution of 4 CAM providers as co-authors in reviewing 2 article on the HPV vaccine. Despite the study may seem innovative from a collaboration point of view, the objective and the methodology do not seem equally worthwhile. Some remarks:

the introduction reports the conclusions (Please see page 2 , lines 68-70). It does not provide the reader with the true objective of the study which is not understandable; the methods are not clear. It is not clear how 4 Swiss CAM providers can represent the international reference standard; the endpoints of the study are unclear; Furthermore, it is not clear whether this collaboration really improves VH because this results is not reported.

All IN ALL, IT IS TOO  little to be disclosed in a peer- reviewed Journal.